# Quantitative Angle Measurement of the Inclined Surface Crack Based on Laser Ultrasonics

**DOI:** 10.3390/s25051486

**Published:** 2025-02-28

**Authors:** Haiyang Li, Rui Zhang, Qianghua Pan, Pengfei Wang

**Affiliations:** 1Shanghai Acoustics Laboratory, Chinese Academy of Sciences, Shanghai 201815, China; lihaiyang@mail.ioa.ac.cn; 2School of Mechanical Science and Engineering, Huazhong University of Science and Technology, Wuhan 430074, China; 3China Special Equipment Inspection and Research Institute, Beijing 100029, China; panqianghua@csei.org.cn; 4School of Information and Communication Engineering, North University of China, Taiyuan 030051, China; pengfei_wang_life@163.com

**Keywords:** laser ultrasonic, surface wave, crack angle measurement

## Abstract

To measure the angles of oblique surface cracks on a specimen, the method of using the time difference between the reflected and scattered waves generated by the interaction of the surface waves and the crack is proposed. Based on the reflection and scattering waves’ paths at the surface crack, an analytical equation that involves the crack depth, width, and inclined angle is developed. On the basis of establishing the ratio of width to depth, ∆t-φ curves and estimated error images without the width term can be analyzed in detail. The finite element simulation and experimental results for aluminum alloy samples show that a crack’s angle can be detected using the time difference method. The effects of the width term in the analytical equation on the estimated angles of the surface cracks are also verified. Measurement of the quantitative angle of inclined surface cracks is first regarded as a multiparameter inversion problem. The simulation and the experimental discussion of the interrelation between the depth, width, and angle of the inclined surface crack in this work are very meaningful for quantitative determination using laser ultrasound.

## 1. Introduction

Surface cracks that are not detected in time will continue to expand due to the external load, causing deterioration in the performance of the part, shortening the service time, and, in severe cases, even causing the part to fracture, finally resulting in great safety hazards. A crack is usually randomly distributed on the surface. Its dimensional parameters, such as its width, depth, and angle, are key parameters directly affecting the speed of the crack’s expansion from a microscopic to a macroscopic crack. Particularly for oblique cracks, the angle of the crack determines the direction of expansion during its growth procedure. So, how to quantitatively measure a crack’s angle is very worthy of deeper research.

Ultrasonic testing (UT) technology uses the transmission, reflection, and scattering phenomena at the discontinuous interfaces generated by cracks to determine the mechanical properties of detected materials. At the early stage, UT can only recognize and locate the existence of large, open cracks according to the wave reflection and transmission at the cracks during wave propagation [1]. After this point, the detection ability of UT is hugely improved, and it can realize micro-crack and closed crack detection thanks to progress in electronic science and technology. To satisfy the ever more precise requirements in the industrial field, a method for quantitively measuring cracks was developed. In addition, another change in UT is the change from the traditional technique of using contact with a specimen via liquid coupling to using non-contact methods suitable for detection under harsh environments. Electromagnetic ultrasound [2], air-coupled ultrasound [3], and laser ultrasonics [4] have emerged as new and non-contact UT methods. The laser ultrasonic technique [5], which does not limit the detected material or the detection distance from the specimen’s surface, was adopted in this work.

Laser ultrasonic inspection technology [6] has wide applications in defect detection [7,8], mechanical parameter measurement [9,10], and structural health monitoring [11] due to advantages such as its wideband, high resolution, non-contact generation, and the different wave types generated in one instance by a laser pump on the surface, including longitudinal waves, shear waves, head waves, and Rayleigh waves. For surface crack detection, Rayleigh waves are chosen as the detection waves on account of their propagation energy being concentrated at two times their wavelength deep under a specimen’s surface, resulting in high sensitivity to surface imperfections. Research focused on Rayleigh waves based on laser ultrasonics has demonstrated that they are an excellent tool for sizing surface cracks. To simplify complex shapes and random distributions, the two kinds of cracks studied in these publications are vertical and oblique cracks. To date, for vertical cracks, crack depth measurement has been the main subject. Li [12] proposed a depth measurement method that adopted a critical frequency for the energy conversion from transmitted waves to reflected waves after the interaction of broadband Rayleigh waves with rectangular cracks. Cooper [13] quantified the crack depth using the time difference between reflected and slot-tip scattered waves to detect surface defects with depths of 0.1 to 0.5 mm. Jeong [14] also conducted an in-depth study of vertical cracks and used Cooper’s method to quantitatively measure cracks with crack depths of 0.3–5 mm. Regarding the research described above, Li’s methods are frequency methods, which are used to extract the frequency characteristics of reflection and transmission acoustic signals to size the crack depth. Cooper’s and Jeong’s methods are time domain methods that make use of the time differences in wave propagation to quantify the crack depths. For oblique cracks, Dutton [15] studied the amplitude and frequency enhancement behavior at oblique cracks using a scanning laser line source or a detector to quantify the angle of the inclined cracks and ultimately found that the scanning detector method was more suitable for detecting oblique cracks. In terms of time domain measurement, Matsuda [16] established an analytical equation by calculating the arrival times of waves along the surfaces of oblique cracks to estimate the angles of these surface cracks. Ni [17] quantified the crack angles based on the arrival times of different waves generated by a pulsed laser beam aimed precisely at the cracks. During the process of measurement, the pulsed laser beam had to be arranged to strike the crack precisely in both Matsuda’s method and Ni’s method. In addition, in both Matsuda’s and Ni’s methods, the accuracy of the measurement was affected if the laser’s position did not accurately irradiate directly above a surface crack. According to the research described above, it can be seen that the time domain method is an effective method for quantifying the depth of vertical cracks and the angles of inclined cracks. However, the current time domain methods for quantifying the angles of inclined cracks require the position of the surface crack to be known in order to direct the incident laser accurately, leading to the failure of these measurement methods when the position of a surface crack is not known. The methods for choosing the scattering waves for the detection of surface cracks and determining their paths are key parts of many time domain methods. In comparison to the other time domain methods, Cooper used a scattering wave generated through mode conversion and reflected waves to size crack depths, and this measurement procedure did not require the position of the surface cracks to be known. Meanwhile, Cooper’s method has not been applied to quantifying the angles of inclined cracks. In addition, all of these publications about sizing surface cracks based on laser ultrasonics regard the quantitative detection of surface cracks as a single-parameter inversion problem. That is to say, the detection of oblique surface crack angles is realized using one parameter, such as the amplitude capture [15] or the time difference in arrival [16,17]. However, surface cracks in the two-dimensional plane have three size parameters, such as depth, width, and an inclined angle. For the quantitative detection of oblique surface crack angles, the effects of the surface crack’s depth and width on the estimated results have not been reported until now. So, the three-parameter inversion problem for sizing oblique surface cracks is a more accurate detection method which will be studied in this work.

In this work, the arrival time difference method is applied to quantifying the angle of oblique surface cracks using laser ultrasonics. The equation based on the wave propagation path contains the three size parameters of the surface cracks, which are the angle, depth, and width, all of which affect each other, allowing one of these three parameters to be measured when the other values are known. The main project of this work is to quantify the angle of oblique surface cracks. This work is mainly divided into four parts: (1) The interaction between direct waves and cracks, producing transmission, reflection, and scattered waves, is theoretically analyzed. (2) A finite element simulation using COMSOL Multiphysics software verifies the applicability of the proposed formulas and analyzes the effect of the surface crack’s width and depth on the accuracy of the angle measurement. (3) An experimental platform for laser ultrasonic inspection is used to study aluminum alloy samples, and the experimental results are analyzed in depth. (4) The main conclusions are discussed in the final part.

## 2. Theoretical Analysis

### 2.1. Analytical Formula for Sizing Angle

An oblique surface crack is a crack starting from the material’s surface and ending at a given location on the material, with a given angle between the crack interface and the material’s surface. The anticlockwise direction is used in this work. The oblique angle of a surface crack reaching from the side surface of a material to the vertical falls within 0° to 90°. An angle of 90° is a special case for an oblique surface crack that is referred to as a vertical crack. Based on the propagation process for acoustic waves and the formula for depth measurements of straight rectangular defects proposed by Cooper [13], counterparts can be obtained for oblique rectangular defects. Figure 1 depicts the acoustic propagation process at an oblique angle defect with the depth d, width w, and inclination angle φ.

The waveform transition phenomenon at the tip of a defect is very complicated. The main waveform in this process involves a reflected sound wave RR, a transmitted sound wave TR, and a scattering wave RS at the tip of the defect, where the surface wave RS is generated after the scattered wave S3 reaches the material’s surface. In Figure 1, the surface wave propagation process is divided into three lines to allow for a clear analysis of this process. The three lines are path 1 (labeled in black), the reflected surface wave RR; path 2 (labeled in blue), the surface wave RT-SR_1_-SR_2_-TR; and path 3 (labeled in red), RT-SR_1_-SR_3_-S_3_-RS. The surface wave R excited by the laser propagates along the material’s surface. When the surface wave encounters the left side of the defect, part of the component is directly reflected by the defect, which directly constitutes the reflected surface wave RR in path 1. Part of the component undergoes a mode shift at the defect boundary vertex, and the resulting longitudinal wave and shear wave propagate to the lower left of the vertex and radiate into the interior of the material. Since this mode transition is not analyzed further in this article, it is not labeled in Figure 1. Another part of the component continues to propagate forward along the defect’s surface to generate the surface wave RT. In the process of propagation, the surface wave RT encounters the tip M on the left side of the bottom of the defect and undergoes the first mode conversion, thereby branching out the common SR_1_ in path 2 and path 3. In this case, the point of impact M is equivalent to a secondary sound source, and most of the energy of the acoustic signal propagating along the defect’s surface radiates to the interior of the solid material as a shear wave signal in the same direction as the original direction of propagation. The remaining part of the acoustic wave’s energy continues to transmit along the bottom of the defect to generate the surface wave SR_1_ in the right direction. When the surface wave SR_1_ encounters point N of the right tip, the second mode conversion occurs, and the surface wave SR_2_ in path 2 and the surface wave SR_3_ in path 3 are separated. The surface wave SR_2_ propagates upward along the right boundary of the defect and becomes the transmitted acoustic wave TR after reaching the material’s surface. The surface wave SR_3_ returns along the bottom of the defect, and the third mode transition occurs at point M. Point M is regarded as another secondary sound field, and most of the energy of SR_3_ enters the material along the bottom of the defect and generates a shear wave S_3_ propagating along θ, which is equal to 30°, and continues to transmit to the material’s surface and is converted into the surface wave RS. The angle θ is a key parameter for calculating the path of the shear wave S_3_, and its value is verified in Appendix A using a simulation method called the mass spring lattice model. The third mode transition also generates a longitudinal wave, which propagates to the lower left of point M and is not transmitted to the material’s surface and converted into a surface wave. The reflected surface wave RR is the surface wave’s acoustic signal directly reflected from the direct surface wave encountering the left surface of the defect in path 1, while the tip-scattered surface wave RS is the surface wave’s acoustic signal generated by the three mode conversions of the surface wave RT in path 3. According to this process, it can be seen that the generation processes for the surface waves RR and RS are very different, which leads to a time delay between these two waves, and this time delay is heavily related to the depth, width, and angle of the oblique rectangular defect. Quantitative detection of a defect’s geometric parameters provides detection methods. In this work, the time delay of the different acoustic signals in path 1 and path 3 is used to measure the depth of the surface defects. An equation for the relation between the angle φ, depth d, and width w is established, as shown in Equation (1):(1)$$∆t=\frac{2v_{r}sin\varphi +2v_{s}sin\left(\theta +\varphi \right)-v_{s}}{2v_{r}v_{s}sin\varphi sin\left(\theta +\varphi \right)}\cdot d+\frac{2w}{v_{r}}$$
where $v_{r}$ and $v_{s}$ are the velocities of the surface wave and shear wave, respectively. Indeed, ∆t is a quantity that can be measured by observing the acquired wave signals, and θ is the scattering angle of the shear wave S_3_ of 30°. So, Equation (1) is an implicit function related to the three size parameters of width w, depth d, and angle φ. The detailed derivation process for (1) can be seen in Appendix B. A three-parameter inversion problem using Equation (1) is proposed in this work. According to a geometric model of a crack, a straight crack can be regarded as an oblique crack with an angle of 90°. Therefore, when φ = 90° is put into the equation, a calculation formula for the depth of a straight crack can be obtained. The result is consistent with [13], which is well verified using the formula. When Jeong [14] measured the depth, he did not consider the influence of width on the measured value, and measurement errors could not be explained reasonably. Therefore, the influence of width and depth on the angle measurement is needed for a further analysis.

### 2.2. The Curve of the Time Difference Versus the Angle

In measuring its angle using Equation (1), the depth d and width w of an oblique surface crack also contribute to the measurement result. However, the effects of depth d and width w are very different. The depth d is directly coupled with the angle of the surface crack, but the width w is a correction item without any coupling with the angle in Equation (1). So, the depth d cannot be ignored during quantitative determination of a surface crack’s angle. The effect of the width w of the surface crack on the angle measurement is an unknown problem to be solved. If the width w of the surface crack is ignored in Equation (1), the calculation formula is transformed as follows:(2)$$\Delta t=\frac{2v_{r}\sin\varphi +2v_{s}\sin\left(\theta +\varphi \right)-v_{s}}{2v_{r}v_{s}\sin\varphi \sin\left(\theta +\varphi \right)}d$$

Comparing Equations (1) and (2), the difference between them only lies in the subsequent term $\frac{2w}{v_{r}}$, which denotes the effect of the surface crack’s width on the arrival time difference. According to Equations (1) and (2), it is seen that the angle φ appears in the form of a trigonometric function, which then theoretically shows a critical value due to |sinφ| ≤ 1. In order to analyze how the arrival time difference ∆t based on Equations (1) and (2) changes with the angle φ, the depth d is set as 1 mm and the width w is set as 0.1 mm, and this crack size is chosen to highlight the relationship between the angle φ and time difference ∆t. The ∆t-φ curve is shown in Figure 2a. Figure 2b shows the first derivatives of the two curves in Figure 2a, and the derivative curves of the two curves coincide as one curve.

According to Figure 2a, it can be observed that with and without the term $\frac{2w}{v_{r}}$, the curves generated using Equations (1) and (2), respectively, follow the same trend in terms of their whole scope. The reason for this is that the term $\frac{2w}{v_{r}}$ is constant, so the trend in the curve ∆t-φ based on Equation (1) does not change without the term $\frac{2w}{v_{r}}$. Moreover, the curve has two parts: one part of the curve is a monotone function when the crack angles are 0–61.35°, and the other part corresponds to when the crack’s tilted angle is larger than 61.35°, and the curve adheres to the phenomenon of decreasing first and then slightly increasing. With the angle ranging within 61.35–90°, there is a pole point in the ∆t-φ curve, which is the minimum of the curve, leading to the two-valued function of the curve in this range.

The time difference ∆t at an angle of 61.35°and at an angle of 90° are the same at 0.585 μs, which can be read directly on the curves. To verify this phenomenon, according to Figure 2b, the first-derivative curve arrives at zero at a surface angle of 75.66°, which is the pole of the curve in Figure 2a, demonstrating that the curve is a two-valued function in the range of 61.35–90°. For this reason, if the time difference in Equation (1) is 0.573–0.585 μs, the angle of the surface crack cannot be measured precisely. So, the resolution range of the measured angle using the proposed method in this work is 0–61.35° for a corresponding time difference within 0.585–1.304 μs.

In this paper, the error caused by ignoring the term $\frac{2w}{v_{r}}$ is defined as(3a)$$Err={Ang}_{w}-Ang$$(3b)$${Err}_{rel}=\frac{{Ang}_{w}-Ang}{{Ang}_{w}}$$
where Ang_w_ and Ang are the estimated angles of the surface crack with and without the term $\frac{2w}{v_{r}}$ when the arrival time difference ∆t is the same value. Err is the absolute error defined by Equation (3a), and Err_rel_ are the relative errors defined by Equation (3b). For example, in Figure 2a, if the arrival time difference ∆t is 0.83 μs, the estimated angle at point A1 is 15.4° according to the curve ∆t-φ without the term $\frac{2w}{v_{r}}$, and the estimated angle at point A2 is 20° according to the curve ∆t-φ with the term $\frac{2w}{v_{r}}$ resulting in the absolute error Err = 4.6°. In the same way, Ang_B1_ = 30.14° and Ang_B2_ = 40° result in the absolute error Err = 9.86°. It can be seen that the absolute error Err increases with the angle of the surface crack by comparing the ∆t-φ curves with and without the term $\frac{2w}{v_{r}}$.

### 2.3. The Influence of Depth and Width on the Estimated Angles

In order to analyze the influence of depth d and width w on the estimated angles in Equations (1) and (2), the relative errors Err_rel_ defined using Equation (3b) are analyzed. Figure 3a shows the relative errors Err_rel_ obtained by changing the width and depth with the crack’s angle fixed at 45°. Figure 3b shows the relative errors Err_rel_ obtained by changing the width and depth with the ratio w/d fixed at 0.2.

It can be ascertained from Figure 3a that the relative errors Err_rel_ increase with the width w and decrease with the depth d of the surface crack. In addition, when the ratio w/d, defined by the width and depth of the surface crack, in Figure 3a is constant, Err_rel_ of the crack angle are constant. To show this phenomenon in detail, the ∆t-φ curves with and without the term $\frac{2w}{v_{r}}$ under the conditions of the depth d being 1 mm and the width being 0.2 mm and the ∆t-φ curves with and without the term $\frac{2w}{v_{r}}$ under the conditions of the depth d being 0.5 mm and the width being 0.1 mm are calculated and presented in Figure 3b. Though the depths and the widths of the surface cracks are different, their w/d ratios are the same at 0.2. The angle at point A2 with the term $\frac{2w}{v_{r}}$ is 40°. That is to say, Ang_w_ = 40° in Equations (3a) and (3b) at point A2. Without the term $\frac{2w}{v_{r}}$, Ang = 23.23° in Equations (3a) and (3b) at point A2. In the same way, Ang_w_ = 40° and Ang = 23.23° in Equations (3a) and (3b) at point B2 and point B1, respectively. It can evidently be seen that both the absolute error Err and the relative errors Err_rel_ are the same at point A2 and point B2 though their depths and widths are different. This is very important for analyzing the effect of the width w and the depth d of a surface crack on the quantitative measurement of its angle. As discussed about Equation (1), the angle inversion problem is an implicit function related to the three size parameters of width w, depth d, and angle φ. Given that Err_rel_ of the crack angle are constant when the ratio w/d of the surface crack is constant, the width w and the depth d of the surface crack are replaced by the w/d ratio when evaluating the angle of the surface crack. On this basis, the three-parameter inversion problem is reduced into a two-parameter problem. A plot of Err_rel_ with the ratio w/d versus the angle is shown in Figure 4.

The pixels in Figure 4 are labeled in different colors that denote the relative errors Err_rel_. The plot of Err_rel_ presents a complex distribution with the ratio w/d and the angle. At the same angle, the relative errors Err_rel_ increase with the ratio w/d. However, at a ratio w/d less than 0.8, Err_rel_ change with the angle. Meanwhile, at some angles, the relative errors Err_rel_ are equal to one. And when the ratio w/d is greater than 0.8, the relative errors Err_rel_ are always close to one. These zones where the relative errors Err_rel_ are equal to one due to Ang in Equation (3b) are set at zero. The reason for Ang = 0° is that the angle measurement method based on Equation (2) does not work because the width w of the surface crack cannot be ignored. To analyze this phenomenon in detail, three dashed lines in black are shown at w/d ratios of 0.5, 0.05, and 3.33, respectively. Their curves ∆t-φ are shown in Figure 5.

In Figure 5a–c, the curve which is calculated using Equation (1) is labeled Line 1, and the curve which is calculated using Equation (2) is labeled Line 2. It can be seen that Line 1 and Line 2 are closest under w/d = 0.05 and the furthest apart from each other under w/d = 3.33. The smallest errors exist in Figure 5b, and the largest errors are shown in Figure 5c. This is consistent with the results in Figure 4. For a ratio w/d = 0.05, the relative errors Err_rel_ are less than 0.35 when the angles range within 0~80°, and Err_rel_ are greater than 0.35 when the angles range within 80~90°. This is because both the ∆t-φ curves based on Equations (1) and (2), respectively, are two-valued functions that have the lowest value at 75.66°. So, when the measured angles are greater than 80°, the estimated errors Err_rel_ are greatly increased. From the view of the dimension parameters of a surface crack, a ratio w/d = 0.05 means that this surface crack has a relatively narrow width or a relatively long depth. If the surface crack’s depth is constant, it is a crack that is narrow enough that the angle measurement based on Equation (2) has low relative errors. This is also consistent with Figure 4. If the ratio w/d is 3.33, the relative errors are equivalent to those in Figure 4. It can be seen in Figure 5c that Line 1 is so far from Line 2 that no estimated angles exist at both Line 1 and Line 2 under the same time difference. From a physical view, it is easy to understand that when the ratio w/d is 3.33, the width is a very large value compared to the depth of the surface crack, meaning that the width w cannot be ignored when quantifying the angle of the surface crack. The two ∆t-φ curves when the w/d ratio is 0.5 are presented in Figure 5a. When the angles range within 0~12.07°, the angle Ang at Line 2 has to be set to zero in Equations (3a) and (3b) due to no angle Ang existing at Line 2 that corresponds to Ang_w_ at Line 1. So, in Figure 4, in this angle range, at the line with a w/d ratio of 0.5, this part of the image is labeled in red due to the relative errors being equal to one. It is worth noting that even when Err_rel_ are one, or 100%, the estimated angle is regarded as 0°. In Figure 4, Err_rel_ of estimated angles from 20° to 70° are a little lower than those of estimated angles from 70° to 90°. The reason for this is that the ∆t-φ curve is a two-valued function, resulting in the estimated angles having larger Err_rel_ after the minimum value. All in all, the ∆t-φ curve is a two-valued function in a given estimated angle range, and this property increases the estimated errors. According to the Err_rel_ plot, the ratio w/d heavily determines the effect of the width on the relative errors Err_rel_. And this effect is also related to the estimated angles. So, the quantitative angles of inclined surface cracks are complex inversion problems.

## 3. Analysis of the FEM Simulation and Results

### 3.1. Simulation Model

Two kinds of excitation sources usually used to generate acoustic waves in specimens are spot sources [18] and line sources [19] according to the type of laser beam focusing during laser ultrasonic detection. For the spot source type, the incident laser is focused as a spot using a lens onto the surface of the specimen, and the Rayleigh wave generated propagates from the spot’s position in all directions. The directivity of the Rayleigh wave generated by a spot laser is weaker than that generated by a line source because the Rayleigh wave generated by a line source only propagates in two opposite directions according to the two sides of the excitation source [20]. So, in this paper, to obtain a vertically incident Rayleigh wave at the surface crack, a line source is used to generate the Rayleigh waves due to the better directivity of their propagation. In this study, in the two-dimensional plane, the line source is described as a product of two Gaussian functions in the spatial and time domains, which is shown as(4)$$s(x,t)=f(x)\cdot g(t)=e^{-\frac{\left(x-x_{0}\right)}{R_{0}^{2}}}\cdot \frac{t}{t_{0}}e^{-\frac{t}{t_{0}}}$$
where $x_{0}$ is the laser operating point, R_0_ is the pulsed laser’s spot radius of 100 μm, and t_0_ is the rise time of the pulse laser at 15 ns. Figure 6 shows the spatial and temporal functions of the line source laser. We set the material as aluminum metal, and the material’s properties are shown in Table 1. Aluminum was chosen as the simulation medium with a Rayleigh wave velocity of 2940 m/s because the same material was also used during the experimental procedure. In this simulation model, the frequency and wavelength of the Rayleigh wave generated by the line source are 1.5 MHz and 1.9 mm, respectively. A rectangular model with a length of 30 mm and a width of 5 mm was meshed using the triangle type with a minimum size of 6 × 10^−4^ mm. The model’s right, left, and lower boundaries were set as low-reflection boundary conditions. A simulation time of 10 μs with a step of 10 ns was adopted in the whole simulation calculation to obtain the acoustic fields generated by the pulsed laser beam.

### 3.2. The FEM Simulation Results

The simulated acoustic fields, including the waves’ path of reflection and the transmission and scattered waves around the surface defect, are presented in Figure 7 for an oblique surface defect set to have a 1.0 mm depth, a 0.5 mm width, and a 60° angle. Four transient moments in the process of wave propagation were chosen, corresponding to the wave paths presented in Figure 7a–d.

Figure 7a shows the first encounter between the incident surface wave R and the defect: a part of the power forms a wave RT propagating along the surface, and the other part forms a reflected wave RR in the opposite direction to the incident wave. The reflected surface wave RR then propagates along the surface and reaches the position in Figure 7c, forming path 1 in Figure 1. The surface wave RT continues to propagate along the sidewall of the defect and reaches the bottom of the defect, and the first mode conversion occurs at point M, resulting in the surface wave SR_1_ and the shear wave component S_1_, as shown in Figure 7b. It can be seen that the surface wave SR_1_ propagates along the bottom of the defect, and the shear wave S_1_ propagates into the material below the bottom of the defect. Wave S_1_ rotates clockwise along the defect’s base by a 30° scattering angle, causing the wave to propagate into the material and not be detected at the specimen’s surface. This not only shows the correctness of the theoretical analysis of Equation (1) but also means that the subsequent surface wave component RS is generated in other processes rather than by the surface wave RT. The surface wave SR_1_ undergoes a second mode transition at point N, resulting in the reflected surface wave SR_3_, as shown in Figure 7c. The Rayleigh wave SR_3_ is divided into three parts, namely the surface wave SR_2_ along the right side of the surface of the crack, the shear wave S_2_ scattered into the material, and the Rayleigh wave SR_3_ reflected along the bottom. The Rayleigh wave SR_3_ continues to propagate and forms the emitted Rayleigh wave RT. This is the wave of path 2 in Figure 1. The surface wave SR_3_ propagates along the bottom of the crack, and when it encounters the tip M, the third mode transition occurs, and a shear component S_3_ is generated at tip M in the material, as shown in Figure 7d. This is path 3 in Figure 1. The wave S_1_ propagates towards the surface along the left surface of the defect with a counterclockwise rotation of 30°. Comparing Figure 7b,d, it can be seen that before the shear wave S_3_ reaches the surface of the sample, no other large-amplitude components are generated on the surface of the material; that is to say, no surface wave RS is generated. Therefore, it can be proven that the surface wave RS is generated by the shear wave S_3_ and not by the surface wave RT. The simulation results obtained using the finite element method verify the theoretical analysis given in Section 2.1.

### 3.3. Simulation Results and Analysis

In order to study the influence of different width w and depth d values on the crack angle measurement, three groups of cracks with different depths and widths are selected. The geometric parameters of cracks that correspond to the w/d ratios at the three black dashed lines in Figure 4 are shown in Table 2.

Based on the simulation setup in Table 2, the acquired acoustic signals in the time domain for samples A, B, and C are shown in Figure 8.

As shown in Figure 8a–c, the wave DR is a direct Rayleigh wave generated using a laser line source. Consistent with the theoretical analysis in Section 2.1 and the simulation results in Section 3.2, the wave RR is a reflected wave at path 1 generated by the interface of the surface crack. The wave RS is a mode-converted wave at the tip of the surface crack which then arrives at the surface of the specimen. The RS wave’s propagation path is path 3. Combined with a B-scan image of the sample, the RS wave’s arrival time is determined precisely. Here, as an example, a B-scan image of sample A at a notch angle of 60° is used, and a value of ∆t of 0.37 μs can be determined from Figure 9.

Under different defect sizes, the arrival time of the surface wave RR is basically unchanged and always remains around 5.5 μs. Given that the surface crack’s angle affects the superposition of the horizontal wave and longitudinal wave, the polarity of the RR wave variates with the oblique angle. For oblique defects with the same depth and width, the defect angle decides the length of path 3 and then affects the propagation time for the RS wave. It can be seen from Figure 8a that the arrival times of the surface wave RS are significantly different. The surface defect angles in the simulation models are estimated using Equations (1) and (2), and the estimated results for samples A, B, and C are shown in Table 3, Table 4 and Table 5.

To show the relative errors for the estimated angles with and without the width term, respectively, more clearly, the estimated results with the width term and without the width term from Table 3, Table 4 and Table 5 are presented in Figure 10.

The ∆t-φ curves at w/d ratios of 0.5, 0.05, and 3.33 for samples A, B, and C with and without the term $\frac{2w}{v_{r}}$ are labeled as Line 1 and Line 2 and are presented in Figure 5. As found in the analyses in Section 2.3, the biggest Err_rel_ are obtained when Line 1 and Line 2 are the farthest apart at a w/d ratio of 3.33, and the lowest Err_rel_ are obtained when they are closest at a w/d ratio of 0.05. Comparing the results in Figure 10a–c, it is seen that the same conclusion can be reached using the simulation data from Table 3, Table 4 and Table 5, verifying the analytical calculation based on Equation (1) for quantifying the angles of surface cracks and the applicable condition for Equation (2) that ignores the width term $\frac{2w}{v_{r}}$.

For the angles estimated with the width term, the two angles estimated correspond to one actual angle during the sizing of the angle of a surface crack in Table 3, Table 4 and Table 5. For example, the angle of 70° for sample A has two estimated angles of 66.59° and 84.75°. The same phenomenon applies to the 90° angle in sample A and the 60°, 70°, and 90° angles for sample B. The reason for this was also discussed—the ∆t-φ curve is a two-valued function in a given angle range where two estimated angles correspond to one actual angle—in Section 2.3. According to Line 1 in Figure 5a, the part of Line 1 where the angles range within 61.35–90° is a two-valued function. The angle of 70° is located within this range, so there are two angles estimated for it. For other angles like the 90° angle for sample A and the 60°, 70°, and 90° angles for sample B, the same explanation works. Moreover, the 80° angle for sample A and the 80° angle for sample B, which are also in the range where the ∆t-φ curve is a two-valued function, only have one estimated angle. Because the arrival time ∆t is 0.33 μs, less than the smallest time difference 0.338 μs on Line 1 in Figure 5a, the 75.66° angle at ∆t = 0.338 μs on Line 1 in Figure 5a is picked as the estimated angle for the 80° angle for sample A. The same processing is applied to the 80° angle for sample B.

For angles estimated without the width term, the estimated angles for sample B are closer than those for sample A to the actual angles. However, for sample C, the measurement method without the width term does not work anymore, leading to all of the estimated angles being zero. From the view of the w/d ratio of the surface crack, the w/d ratios of sample A and sample B are lower that that of sample C; however, the w/d ratio for sample B is even smaller that for sample A, and the best-estimated results are obtained for sample B. That is to say, the width is such a small value compared to the depth of the surface crack in sample B that the width term can be ignored with few errors. This is consistent with the analyses in Section 2.3. For sample C, the reason why the results estimated without the width term are zero is also presented in Section 2.3. Comparing Line 1 and Line 2 in Figure 5c, no angle estimated without the width term on Line 2 corresponds to the angle calculated using the width term on Line 1. So, the angles estimated without the width term are set to zero, leading to the relative errors Err_rel_ being one, presented in Figure 4 in red, or 100%, as presented in Table 5. In this case, the width of the surface crack cannot be ignored when measuring the angle of the surface crack, or huge estimation errors will be obtained.

In addition, for the angle measurement method using Equation (1) with the width term, comparing the estimated results for samples A, B, and C, better results are obtained for samples A and B than sample C. This is because the defect’s width is relatively large, and too long an acoustic path will attenuate the amplitude of the SR_1_ and SR_3_ waves. As a result, the arrival time of the surface wave RS becomes inaccurate. This result is consistent with the experimental results in [13].

Above all, two points based on Equations (1) and (2) are validated here using the simulation results. One of them is that the w/d ratio of the surface crack hugely affects the accuracy of the estimated angle, meaning that quantitative angle determination for surface cracks is a multiparameter inversion problem. The other point is that the ∆t-φ curve is a two-valued function, causing a poor evaluation result to be generated in the two-valued range.

## 4. Experimental Procedures

### 4.1. Experimental Measurement Design

The laser ultrasonic inspection platform used in this paper is a CFR200 laser generator (Quantel Laser of Lumibird, French) as the laser excitation part and a QUARTER-500 mV laser ultrasonic receiver based on the Michelson interferometer principle as the receiving part. Both are produced by BOSS ANOVA Technology Company in the United States. The light emitted from the excitation section passes through a cylindrical lens and forms a line source on the sample’s surface to excite a surface acoustic wave, and the echo acoustic signals are ultimately received by the laser interferometer part. The laser generator’s laser wavelength is 1064 nm, its pulse width is 11 ns, and its pulse repetition frequency is 20 Hz. The laser ultrasonic receiver’s detection sensitivity is 1 × 10^−5^ nm/Hz^1/2^, the laser wavelength is 532 nm, and the bandwidth is 100 kHz~20 MHz. The laser ultrasonic detection platform is equipped with an automatic sweeping frame, which can complete the A-scan and the B-scan imaging to detect the samples. Horizontal scanning with the scanning step size set to 0.12 mm and a scanning distance of 15 mm is performed using the LU Scan software. The experimental samples are five aluminum alloy rectangular plates with dimensions of 100 mm × 30 mm × 8, and artificial rectangular notches are made at 30 mm. In aluminum, the longitudinal wave velocity is 6.440 × 10^3^ m/s, the shear wave velocity is 3.090 × 10^3^ m/s, and the surface wave velocity is 2.940 × 10^3^ m/s. The pulsed laser’s spot radius is 700 μm, and the Rayleigh wave generated in the sample has a center frequency of 1.75 MHz and a wavelength of 1.68 mm. The experimental platform and samples are shown in Figure 11 [21]. Surface cracks were formed on the samples using wire cutting according to the People’s Republic of China machinery industry standard “JBT8428-2006 The Non-destructive testing Blocks for ultrasonic testing” [22].

### 4.2. Theoretical ∆t-φ Curve Analysis

The crack size parameters of the experimental samples were as follows: the width was 0.2 mm, the depth was 0.5 mm, and the angles were within 30–80°. The experiments were conducted on the experimental platform built in Section 4.1. The w/d ratio of the experimental samples was 0.4. Selecting a crack with a 0.2 mm width and a 0.5 mm depth avoided attenuation of the amplitude of the SR_1_ and SR_3_ waves due to an excessively long acoustic diameter, which would have resulted in the arrival time of the scattered wave RS becoming inaccurate. Theoretical ∆t-φ curves with and without the width term, respectively, were drawn and are shown in Figure 12.

Just as elucidated in the analyses in Section 2.3, as can be seen from Figure 12, the arrival time difference Δt ranges within 0.494–0.754 μs on Line 1, and the estimation method using Equation (4) does not work, meaning that there is no angle on Line 2 that corresponds to that on Line 1. The arrival time difference Δt ranges within 0.388–0.395 μs, and Line 1 is a two-valued function, leading to a bigger error being generated in the angles estimated in this range. The white line in Figure 4 corresponds to the experimental sample whose w/d ratio was 0.4. According to the Err_rel_ plot in Figure 4, angle measurement based on Equations (1) and (2) when the angle ranges within 20–60° has good accuracy. It is very helpful to analyze the estimated angles based on the experimental data using the analytical curve in Figure 12.

### 4.3. B-Scan Images of the Surface Cracks

B-scan images acquired by the experimental platform from Figure 11 are shown in Figure 13a–e, respectively.

As shown in Figure 13a, the reflected wave RR generated at path 1 and the scattered wave RS obtained at path 2 are observed, and their paths are labeled with straight dashed lines. It can evidently be seen that their paths are parallel on account of the propagation of the arrival time difference ∆t between the RS and RR waves only being related to the dimension parameters of the surface crack and having nothing to do with the wave propagation distance. The other waves generated at the surface crack are also acquired and labeled, such as the direct wave and the TR wave. The arrival time difference Δt for the RR wave and the RS wave on the experimental samples are selected for calculating the angles using Equations (1) and (2).

### 4.4. Analysis of the Experimental Results

The acoustic waves in the time domain selected from the B-scan images in Figure 13 are shown in Figure 14.

It is seen in Figure 14 that the RR wave and RS wave are two adjacent waves, and the difference in the arrival time between them decreases as the angle decreases. In Figure 14, it is seen that the arrival time of the reflected RR and scattered RS waves is extracted, and the evaluated angles are calculated using Equations (1) and (2) and are shown in Table 6.

To more clearly present the data in Table 6, the angles estimated with and without the width term are presented in Figure 15.

According to Figure 12, Line 1 has two parts: one of them is a two-valued function within the range of 61.35~90°, and the other is a monotone function within the range of 0~61.35°. The 30°, 45°, and 60° angles on the experimental samples are located in the range of 0~61.35°, and the angles estimated using the width item are straightforwardly obtained by inputting the arrival time difference ∆t into Equation (1). For the 70° and 80° angles, which are within the range of 61.35~90° where the ∆t-φ curve is a two-valued function, on the experimental samples, their arrival time differences ∆t are the same at 0.38 μs, being less than 0.388 μs at the lowest point, so the angle estimated with the width term for both the 70° and 80° angles is 75.66°. Moreover, Line 2 is fairly far from Line 1 in Figure 12. The relative errors Err_rel_ are 0.654 at a w/d ratio = 0.4 in Figure 5. It is seen that the relative errors based on the angles estimated without the width term are greater than 60%. Due to the w/d ratio being 0.4, the width is not a low value relative to the depth of the surface crack, so the estimated errors generated by ignoring the width of the surface crack for angle detection are non-negligible. So, the experimental results verify the analyses in Section 2, as well as the simulation results in Section 3.

## 5. Conclusions

A method and formula for the quantitative measurement of the depth and angles of cracks are proposed according to the modal transition phenomenon of the interaction between surface acoustic waves generated by laser ultrasound and inclined cracks. This theory is validated using finite element simulations and experimental studies.

(1) Using the time difference between the scattered wave and the reflected wave reaching a detection point, a measurement formula including the angle, depth, and width, with the third parameter measured by the other two parameters, is established.

(2) Based on theory, the effects of the depth d term and the width w term in the formula on the measured angle are analyzed, and it is concluded that the depth term is directly coupled with the angle and cannot be ignored, while the width term is not directly related to the angle. Therefore, whether the width term can be ignored is discussed. To fix the angle, a relationship diagram for the width, depth, and Err_rel_ is established. According to the relationship diagram, Err_rel_ is related to the ratio of width to depth, w/d.

(3) Several sets of samples with different w/d ratios were used, combining the finite element simulation method with experiments, which were consistent with the theoretical results. When w/d is 0.05, the relative errors Err_rel_ are low. When w/d is between 0.05 and 0.8, the relative errors Err_rel_ change with the angle. When the w/d ratio is greater than 0.8, the angle can hardly be measured if ignoring the width term w.

## Figures and Tables

**Figure 1 sensors-25-01486-f001:**
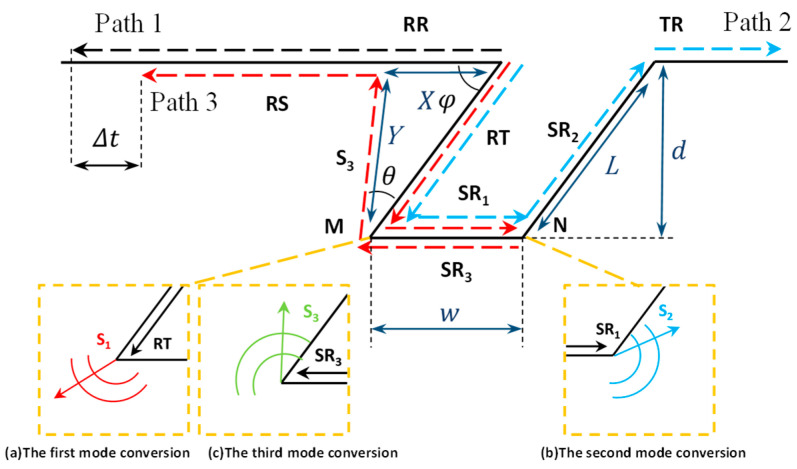
The waveform conversion phenomenon at the tip of a surface defect.

**Figure 2 sensors-25-01486-f002:**
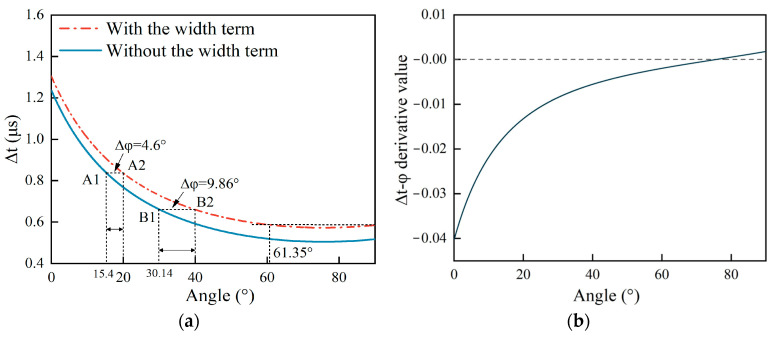
(**a**) The curve ∆t-φ based on Equations (1) and (2). (**b**) Equations (1) and (2) first-derivative curve with respect to φ.

**Figure 3 sensors-25-01486-f003:**
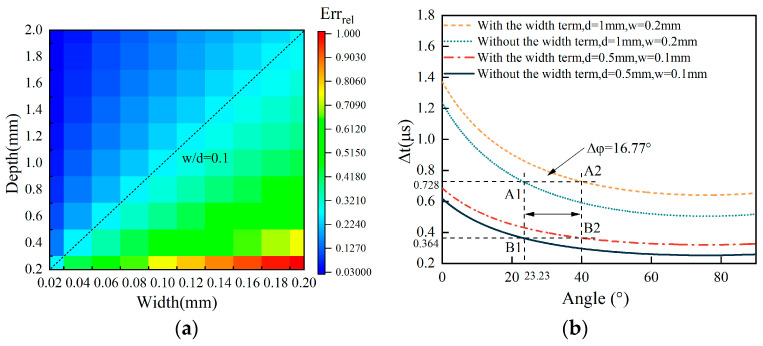
(**a**) The relative errors Err_rel_ obtained with the crack angle fixed at 45°. (**b**) The relative errors Err_rel_ obtained with the ratio w/d fixed at 0.2.

**Figure 4 sensors-25-01486-f004:**
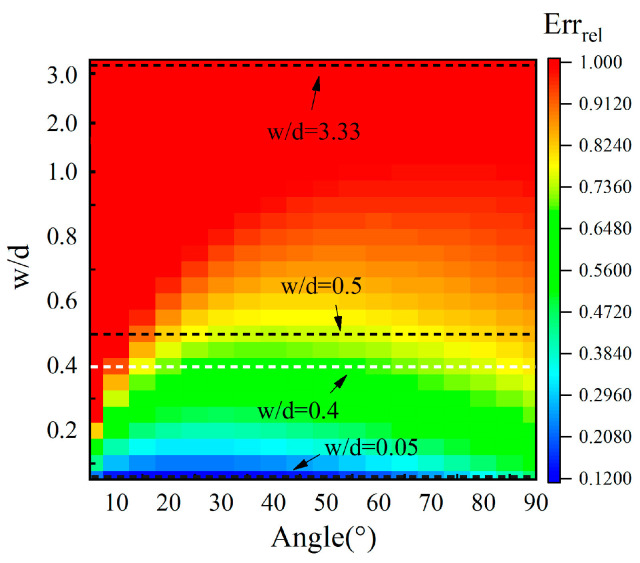
The relative errors Err_rel_ by the ratio w/d versus the angle.

**Figure 5 sensors-25-01486-f005:**
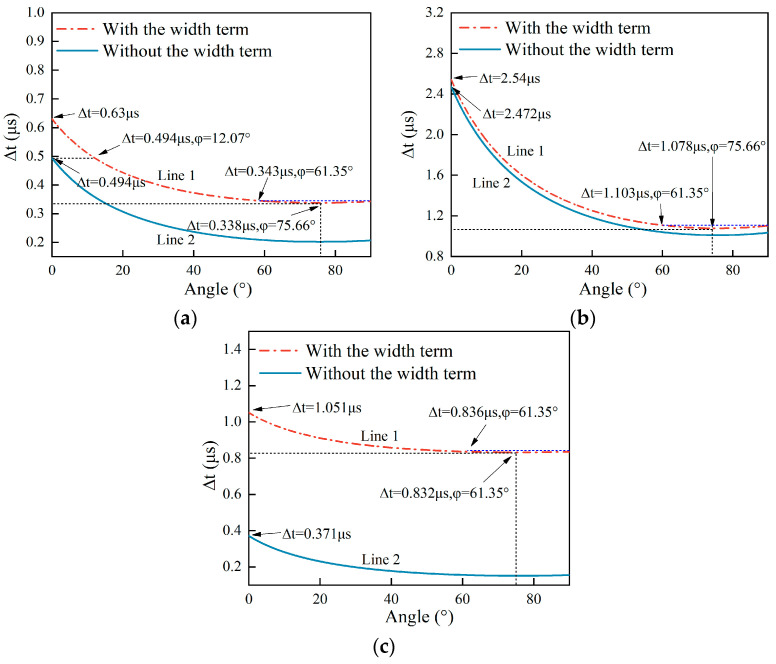
(**a**) The ∆t-φ curve based on Equations (1) and (2) at a ratio w/d = 0.5. (**b**) The ∆t-φ curve based on Equations (1) and (2) at a ratio w/d = 0.05. (**c**) The ∆t-φ curve based on Equations (1) and (2) at a ratio w/d = 3.33.

**Figure 6 sensors-25-01486-f006:**
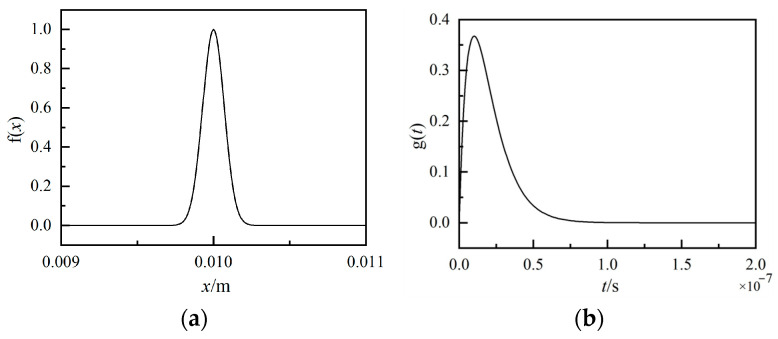
Linear source laser function. (**a**) Spatial function. (**b**) Time function.

**Figure 7 sensors-25-01486-f007:**
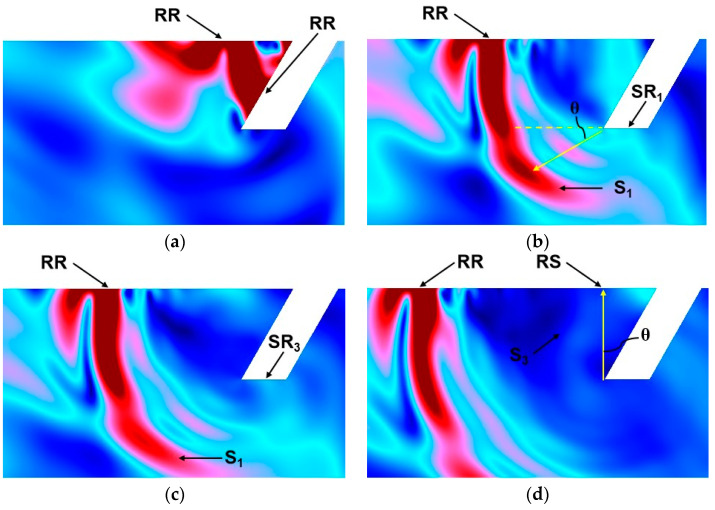
The acoustic field around the notch at (**a**) 3.91 µs. (**b**) 4.38 µs. (**c**) 4.47 µs. (**d**) 4.66 µs.

**Figure 8 sensors-25-01486-f008:**
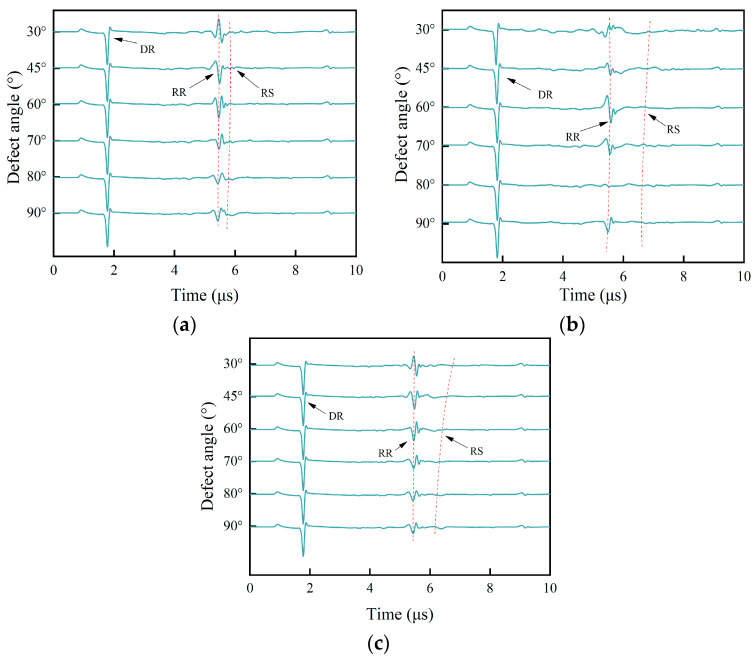
(**a**) Sample A’s signals in the time domain. (**b**) Sample B’s signals in the time domain. (**c**) Sample C’s signals in the time domain.

**Figure 9 sensors-25-01486-f009:**
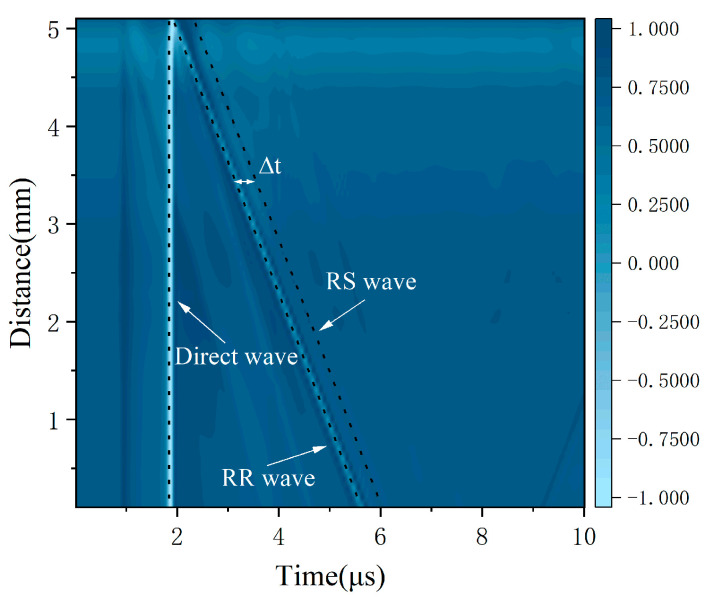
B-scan images of surface cracks (the amplitude is normalized).

**Figure 10 sensors-25-01486-f010:**
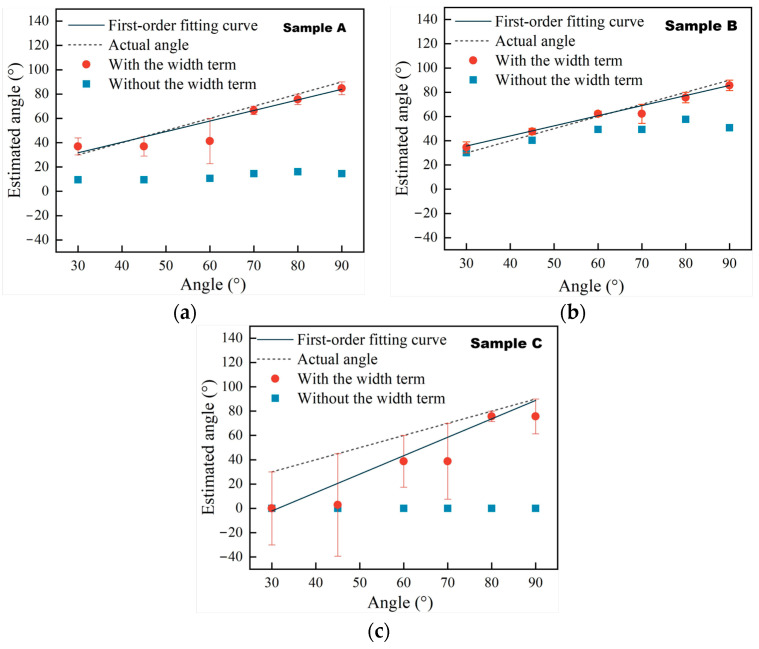
(**a**) Estimated angle and error of sample A. (**b**) Estimated angle and error of sample B. (**c**) Estimated angle and error of sample C.

**Figure 11 sensors-25-01486-f011:**
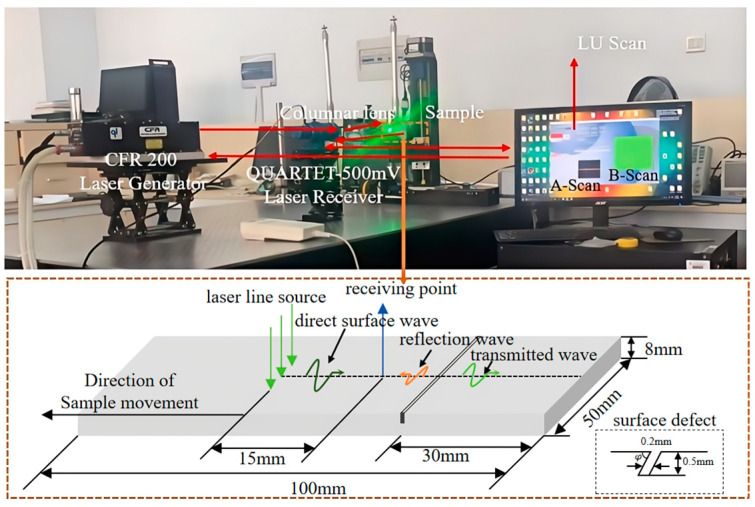
Laser ultrasonic detection system and samples.

**Figure 12 sensors-25-01486-f012:**
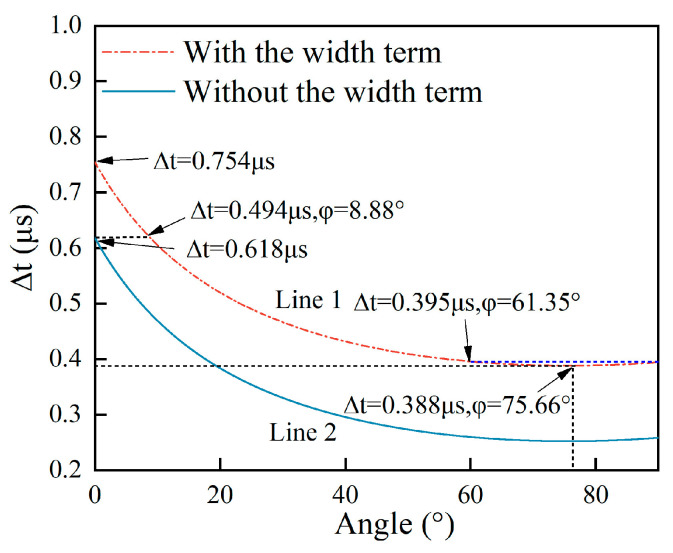
The ∆t-φ curves based on Equations (1) and (2).

**Figure 13 sensors-25-01486-f013:**
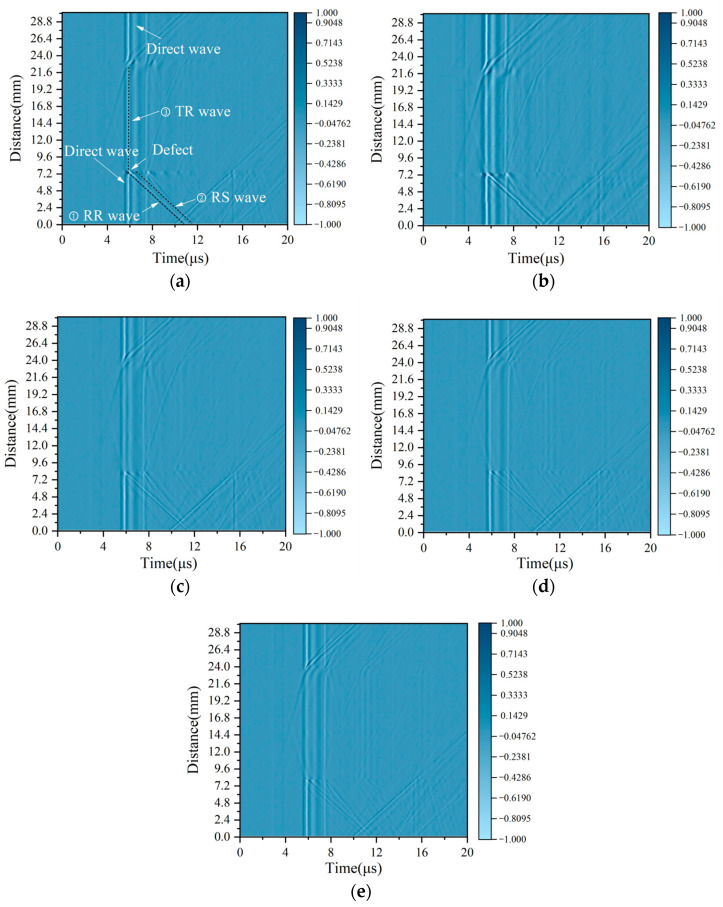
B-scan images of surface cracks (the amplitude is normalized) (**a**) The notch angle is 30°. (**b**) The notch angle is 45°. (**c**) The notch angle is 60°. (**d**) The notch angle is 70°. (**e**) The notch angle is 80°.

**Figure 14 sensors-25-01486-f014:**
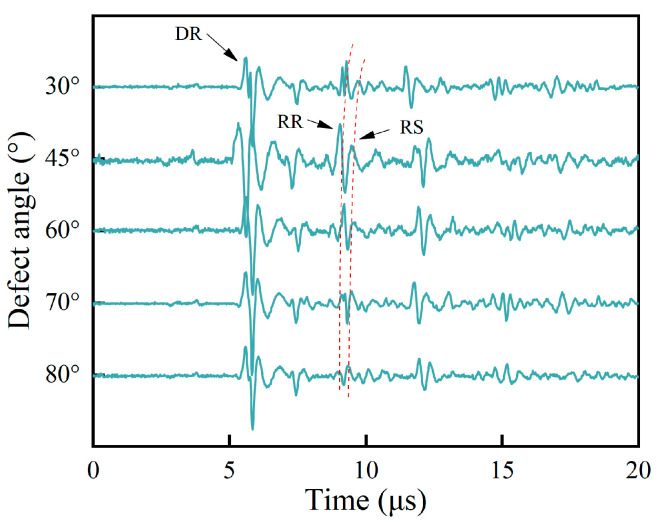
Signals in the time domain.

**Figure 15 sensors-25-01486-f015:**
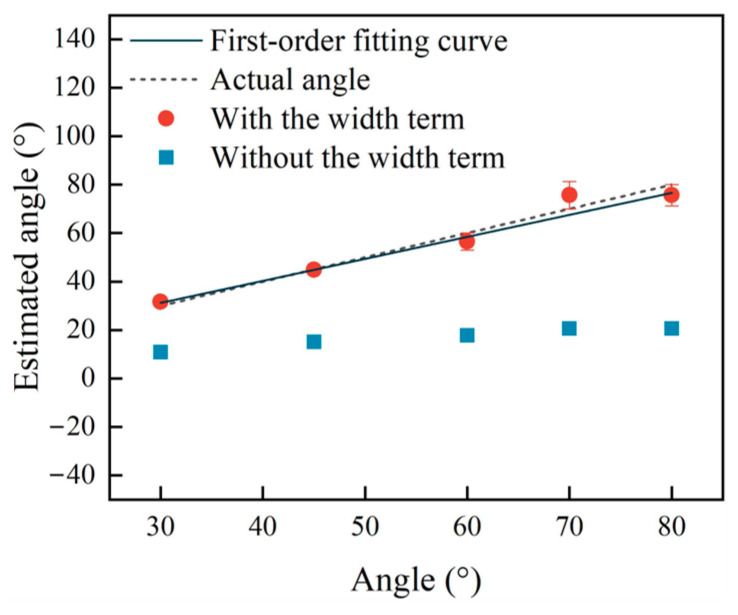
Estimated angles and errors.

**Table 1 sensors-25-01486-t001:** The simulation parameters of the aluminum model.

Parameters	Values
Longitudinal wave	6300 m/s
Shear wave	3090 m/s
Rayleigh wave velocity	2940 m/s
Heat conductivity coefficient	238 W/(m·K)
Thermal expansion coefficient	23 × 10^−6^ 1/K

**Table 2 sensors-25-01486-t002:** Defect size.

Sample	Angle/°	Depth/mm	Width/mm	w/d
A	30–90	0.4	0.2	0.5
B	30–90	2	0.1	0.05
C	30–90	0.3	1	3.33

**Table 3 sensors-25-01486-t003:** Estimated crack angles and relative errors of sample A.

Angle/°	RR/µs	RS/µs	∆t/µs	With the Width Item/°	Relative Error	Without the Width Item/°	Relative Erro
30	5.46	5.84	0.38	36.98	23.27%	9.48	83.82%
45	5.5	5.88	0.38	36.98	17.82%	9.48	68.4%
60	5.47	5.84	0.37	41.32	31.13%	10.63	82.28%
70	5.48	5.82	0.34	66.59/84.75	4.87%/21.07%	14.56	78.07%
80	5.43	5.76	0.33	75.66	5.43%	16.07	79.2%
90	5.43	5.77	0.34	66.59/84.75	14.9%/5.83%	14.56	79.91%

**Table 4 sensors-25-01486-t004:** Estimated crack angles and relative errors of sample B.

Angle/°	RR/µs	RS/µs	∆t/µs	With the Width Item/°	Relative Error	Without the Width Item/°	Relative Error
30	5.53	6.85	1.32	34.56	15.2%	30.14	0.47%
45	5.57	6.75	1.18	47.64	5.87%	40.36	10.31%
60	5.58	6.68	1.1	62.2/89.15	3.67%/48.58%	49.22	17.97%
70	5.56	6.66	1.1	62.2/89.15	11.14%/27.36%	49.22	29.69%
80	5.53	6.58	1.05	75.66	5.43%	57.67	27.91%
90	5.49	6.58	1.09	65.64/85.71	27.07%/4.77%	50.64	43.73%

**Table 5 sensors-25-01486-t005:** Estimated crack angles and relative errors of sample C.

Angle/°	RR/µs	RS/µs	∆t/µs	With the Width Item/°	Relative Error	Without the Width Item/°	Relative Error
30	5.46	6.78	1.32	0	100%	0	100%
45	5.48	6.5	1.02	2.83	93.71%	0	100%
60	5.46	6.32	0.86	38.75	35.42%	0	100%
70	5.45	6.31	0.86	38.75	44.64%	0	100%
80	5.43	6.23	0.8	75.66	5.43%	0	100%
90	5.45	6.17	0.72	75.66	15.93%	0	100%

**Table 6 sensors-25-01486-t006:** Estimated crack angles and relative errors.

Angle/°	RR/µs	RS/µs	∆t/µs	With the Width Item/°	Relative Error	Without the Width Item/°	Relative Error
30	9.28	9.74	0.46	31.62	5.4%	10.87	63.77%
45	9.04	9.46	0.42	44.82	0.4%	15.15	66.33%
60	9.2	9.6	0.4	56.46	5.9%	17.7	70.5%
70	9.1	9.48	0.38	75.66	8.09%	20.6	70.57%
80	8.96	9.34	0.38	75.66	5.43%	20.6	74.25%

## Data Availability

The data that support the findings of this study are available from the corresponding author, Haiyang Li, upon reasonable request.

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
