# Peer review of "Quantitative Angle Measurement of the Inclined Surface Crack Based on Laser Ultrasonics"

_sensors, 2025, doi:10.3390/s25051486_

Round 1

Reviewer 1 Report

Comments and Suggestions for Authors

The mathematical analysis of laser surface wave monitoring for oblique cracks is carried out in this work. The work is very innovative. However, some changes are still needed before publication.

1、The paper research on laser ultrasound crack monitoring can also be appropriately supplemented as follows.

Yang, Dingmin, et al. "Multiple domain dynamic feature adaption transfer learning method for stranded wires health monitoring under variable vibration working conditions using laser-generated ultrasonic guided wave." Engineering Structures 297 (2023): 117013.

2、The order of the graph is a bit messy, and some of the graph is beyond the text scope.

3、The format of the equation is not uniform.

Comments on the Quality of English Language

The English expression of this article needs improvement. Such as, this article uses too many long sentences, and there are too many prepositions in the sentence.

Reviewer 2 Report

Comments and Suggestions for Authors

This paper proposed a time difference method to realize Quantitative angle measurement of the inclined surface crack and built an analytical equation that involves surface crack depth, width and inclined angle. The crack angles of the finite element simulation and experimental results for the aluminum alloy samples were detected via the time difference method. On the basis of a careful review of the manuscript, it cannot be accepted for publication in sensors. in the present form. A major revision is essentially required due to the following comments and concerns.

(1) There are two formula (2), the first formula (2) should be formula (1), and the symbol q in formula (1) wasn’t illustrated. Formulas (B3) and (B4) are the same with those (B1-1) and (B1-2).

(2) In the time difference method proposed in this paper, there are three parameters including crack depth, width and inclined angle. The detected accuracy seems higher than the single crack parameter inversion problem. However, this paper did not provide an efficient method to solve this three parameters inversion problem numerically or experimentally except the results for inclined angle with the depth and width fixed. So please state the major novelties of the current manuscript, and elaborate how to detect the depth, width and inclined angle of a crack with the proposed method.

(3) Please compare the experimental results of detected angles based on time difference method with the real crack data of the experiment samples and discuss the applicability of the analytical equation (formula (1)) in time difference method.

Reviewer 3 Report

Comments and Suggestions for Authors

This paper proposes an innovative method for detecting surface crack geometries using Rayleigh waves generated by laser ultrasonics. By analyzing the time difference between reflected and scattered waves at the surface cracks, the authors derive analytical equations that account for crack depth, width and inclination angle. The proposed methodology and the accuracy of the angle estimation are discussed on the basis of finite element simulations and experimental results, with particular attention to the impact of crack width on the angle measurement. The following key points need to be addressed before publication of the work.

1、The determination of the crack angle is based on the measurement of the arrival time of the wave RS. The time traces and simulation results presented in the manuscript do not clearly show this pulse. The robustness and limitations of results are restricted from simulation and experimental data. So what is the practical significance of proposing the use of laser ultrasonics to detect the depth of surface cracks on metal?

2、The model relies heavily on scattering at points M and N, assuming that these points are acute angles. Is this the case in the measurements? How did you machine the inclined slot in the aluminium samples?

3、The illustration of the wave scattering (Figure 8) does not show the different types of waves scattered at the crack (too small amplitudes compared with the reflected wave RR).

4Change the equation number (2) on page 4 to (1).

5Remove the unnecessary period at the end of the caption for Figure A1. Your manuscript needs careful editing and particular attention to English grammar, spelling, and sentence structureas well as other writing details

Round 2

Reviewer 2 Report

Comments and Suggestions for Authors

Appreciated efforts for this work.